# SIMLR: Machine Learning inside the SIR Model for COVID-19 Forecasting

Roberto Vega [1,2,*] , Leonardo Flores [3] and Russell Greiner [1,2]

1   Department of Computing Science, University of Alberta, Edmonton, AB T6G 2R3, Canada;
    rgreiner@ualberta.ca
2   Alberta Machine Intelligence Institute, Edmonton, AB T5J 3B1, Canada
3   Independent Researcher, San Luis Potosi 78170, Mexico; leonardo.flores.q@gmail.com
*   Correspondence: rvega@ualberta.ca

**Abstract:** Accurate forecasts of the number of newly infected people during an epidemic are critical for making effective timely decisions. This paper addresses this challenge using the SIMLR model, which incorporates machine learning (ML) into the epidemiological SIR model. For each region, SIMLR tracks the changes in the policies implemented at the government level, which it uses to estimate the time-varying parameters of an SIR model for forecasting the number of new infections one to four weeks in advance. It also forecasts the probability of changes in those government policies at each of these future times, which is essential for the longer-range forecasts. We applied SIMLR to data from in Canada and the United States, and show that its mean average percentage error is as good as state-of-the-art forecasting models, with the added advantage of being an interpretable model. We expect that this approach will be useful not only for forecasting COVID-19 infections, but also in predicting the evolution of other infectious diseases.

**Keywords:** COVID-19; probabilistic graphical models; interpretable machine learning

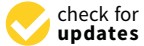



## 1. Introduction

Since its identification in December 2019, COVID-19 has posed critical challenges for the public health and economies of essentially every country in the world [1–3]. Government officials have taken a wide range of measures in an effort to contain this pandemic, including closing schools and workplaces, setting restrictions on air travel, and establishing stay at home requirements [4]. Accurately forecasting the number of new infected people in the short and medium term is critical for the timely decisions about policies and for the proper allocation of medical resources [5,6].

There are three basic approaches for predicting the dynamics of an epidemic: compartmental models, statistical methods, and ML-based methods [5,7]. Compartmental models subdivide a population into mutually exclusive categories, with a set of dynamical equations that explain the transitions among categories [8]. The Susceptible-Infected-Removed (SIR) model [9] is a common choice for the modelling of infectious diseases. Statistical methods extract general statistics from the data to fit mathematical models that explain the evolution of the epidemic [6]. Finally, ML-based methods use machine learning algorithms to analyze historical data and find patterns that lead to accurate predictions of the number of new infected people [7,10].

Arguably, when any approach is used to make high-stake decisions, it is important that it be not just accurate, but also interpretable: It should give the decision-maker enough information to justify the recommendation [11]. Here, we propose SIMLR, which is an interpretable probabilistic graphical model (PGM) that combines compartmental models and ML-based methods. As its name suggests, it incorporates machine learning (ML) within an SIR model. This combines the strength of curve fitting models that allow accurate predictions in the short-term, involving many features, with mechanistic models that allow to extend the range to predictions in the medium and long terms [12].

SIMLR uses a mixture of experts approach [13], where the contribution of each expert to the final forecast depends on the changes in the government policies implemented at various earlier time points. When there is no recent change in policies (two to four weeks before the week to be predicted), SIMLR relies on an SIR model with time-varying parameters that are fitted using machine learning methods. When a change in policy occurs, SIMLR instead relies on a simpler model that predicts that the new number of infections will remain constant. Note that forecasting the number of new infections one and two weeks in advance ($\Delta I_1$ and $\Delta I_2$) is relatively easy as SIMLR knows, at the time of the prediction, whether the policy has changed recently. However, for three- or four-week forecasts ($\Delta I_3$ and $\Delta I_4$), our model needs to estimate the likelihood of a future change of policy. SIMLR incorporates prior domain knowledge to estimate such policy-change probabilities.

The use of such prior models—here epidemiological models—is particularly important when the available data is scarce [14]. At the same time, machine learning models need to acknowledge that the reported data on COVID-19 is imperfect [15,16]. The use of probabilistic graphical models allows SIMLR to account for this uncertainty on the data. At the same time, the probability tables associated with this graphical model can be manually modified to adapt SIMLR to the specific characteristics of a region.

This work makes three important contributions. (1) It empirically shows that an SIR model with time-varying parameters can describe the complex dynamics of COVID-19. (2) It describes an interpretable model that predicts the new number of infections one to four weeks in advance, achieving state-of-the-art results, in terms of mean absolute percentage error (MAPE), on data from Canada and the United States. (3) It presents a machine learning model that incorporates the uncertainty of the input data and can be tailored to the specific situations of a particular region.

The rest of Section 1 describes the related work and the basics of the SIR compartmental model. Section 2 then describes in detail our proposed SIMLR approach. Section 3 shows the results of the predicting the number of new infections in the United States and provinces of Canada. Finally, Section 4 presents our final remarks.

### 1.1. Basic SIR Model

The Susceptible-Infected-Removed (SIR) compartmental model [9] is a mathematical model of infectious disease dynamics that divide the population into three disjoint groups [8]. Susceptible (S) refers to the set of people who have never been infected but can acquire the disease. Infected (I) refers to the set of people who have and can transmit the infection. Removed (R) refers to the people who have either recovered or died from the infection and cannot transmit the disease anymore. This model is defined by the differential equations:

$$\frac{dS}{dt} = -\frac{\beta S(t)I(t)}{N}, \quad \frac{dI}{dt} = \frac{\beta S(t)I(t)}{N} - \gamma I(t), \quad \frac{dR}{dt} = \gamma I(t) \tag{1}$$

SIR assumes an homogeneous and constant population, and it is fully defined by the parameters $\beta$ (transmission rate) and $\gamma$ (recovery rate). The intuition behind this model is that every infected patient gets in contact with $\beta$ people. Since only the susceptible people can become infected, the chance of interacting with a susceptible person is simply the proportion of susceptible people in the entire population, $N = S + I + R$. Likewise, at every time point, $\gamma$ proportion of the infected people is removed from the system. Figure 1a depicts the general behaviour of an SIR model.

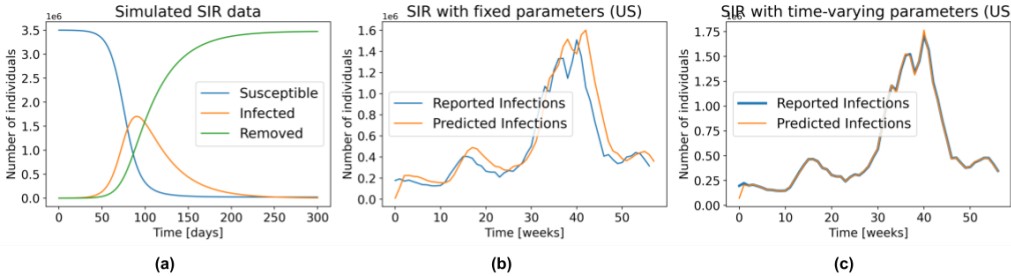

**Figure 1.** (**a**) General behaviour of the SIR model. (**b**) The number of infections predicted by the SIR model with fixed parameters, fitted to the US data for 1 week prediction. (**c**) Similar to (**b**), but with time-varying parameters.

### 1.2. Related Work

The main idea behind combining compartmental models with machine learning is to replace the fixed parameters of the former with time-varying parameters that can be learned from data [6,17–19]. However, most of the approaches focus on finding the parameters that can explain the past data, and not on predicting the number of newly infected people. Although those approaches are useful for obtaining insight into the dynamics of the disease, it does not mean that those parameters will accurately predict the behaviour in the future.

Particularly relevant to our approach is the work by Arik et al. [5], who used latent variables and autoencoders to model extra compartments in an extended Susceptible-Exposed-Infected-Removed (SEIR) model. Those additional compartments bring further insight into how the disease impacts the population [20,21]; however, our experiments suggest that they are not needed for an accurate prediction of the number of new infections. One limitation of their model is a decrease in performance when the trend in the number of new infections changes. We hypothesize that those changes in trend are related to the government policies that are in place at a specific point in time. SIMLR is able to capture those changes by tracking the policies implemented at the government level.

A different line of work replaces epidemiological models with machine learning methods to directly predict the number of new infections [22–25]. Importantly, Yeung et al. [26] added non-pharmaceutical interventions (policies) as features in their models; however, their approach is limited to make predictions up to two weeks in advance, since information about the policies that will be implemented in the future is not available at inference time. Our SIMLR approach differs by being interpretable and also by forecasting policy changes, which allows it to extend the horizon of the $\Delta I$ predictions.

There are many models that attempt to predict the evolution of the COVID-19 epidemic. The Center for Disease Control and Prevention (CDC) in the United States allows different research teams across the globe to submit their forecasts of the number of cases and deaths 1 to 8 weeks in advance [27]. More than 100 teams have submitted at least one prediction to this competition. We compare SIMLR with all of the models that made predictions 1 to 4 weeks in advance in the same time span as our study.

## 2. Materials and Methods

We view SIMLR as a probabilistic graphical model that uses a mixture of experts approach to forecast the number of new COVID-19 infections, 1 to 4 weeks in advance. Figure 2 shows the intuition behind SIMLR. Changes in the government policies are likely to modify the trend of the number of new infections. We assume that stronger policies are likely to decrease the number of new infections, while the opposite effect is likely to occur when relaxing the policies. These changes are reflected as a change in the parameters of the SIR model. Using those parameters, we can then predict the number of new infections, then use that to compute the likelihood of observing other new policy changes in the short term.

While Figure 2 is an schematic diagram used for pedagogical purposes; Figure 3 depicts the formal probabilistic graphical model, as a plate model, that we use to estimate the parameters of the SIR model, the number of new infections, and the likelihood of

observing changes in policies 1 to 4 weeks in advance. The blue nodes are estimated at every time point, while the values of the green nodes are either known as part of the historical data, or inferred in a previous time point. The random variables are assumed to have the following distributions:

$$
\begin{aligned}
\mathrm{CT}_{t+1} \mid & \{\mathrm{CP}_{t-\tau}\}_{\tau \in \{1,2,3\}} & \sim & \quad Cat_{K \in \{-1,0,1\}}(\theta_{CT}) \\
\beta_{t+1} \mid \{\beta_{t-\tau}\}_{\tau \in \{0,1,2\}}, & \mathrm{CT}_{t+1} & \sim & \quad \mathcal{N}(\mu_\beta, \Sigma_\beta) \\
\gamma_{t+1} \mid \{\gamma_{t-\tau}\}_{\tau \in \{0,1,2\}}, & \mathrm{CT}_{t+1} & \sim & \quad \mathcal{N}(\mu_\gamma, \Sigma_\gamma) \\
\mathrm{SIR}_{t+1} \mid & \beta_{t+1}, \gamma_{t+1} & \sim & \quad \mathcal{N}(\mu_{SIR}, \Sigma_{SIR}) \\
U_t \mid & \{\mathrm{SIR}_{t-\tau}\}_{\tau \in \{0,1,2\}} & \sim & \quad Cat_{K \in \{-1,0,1\}}(\theta_U) \\
O_t \mid & W_t & \sim & \quad Cat_{K \in \{0,1\}}(\theta_O) \\
\mathrm{CP}_{t+1} \mid & O_t, U_t & \sim & \quad Cat_{K \in \{-1,0,1\}}(\theta_{CP})
\end{aligned}
\tag{2}
$$

where $t$ indexes the current week, $SIR_t = [S_t, I_t, R_t]$, $\mu_{SIR} \in \mathbb{R}^3$ is given below by Equation (3), $\mu_\beta = (\alpha_{0,CT_{t+1}}) + (\alpha_{1,CT_{t+1}})\beta_{t-1} + (\alpha_{2,CT_{t+1}})\beta_{t-2} + (\alpha_{3,CT_{t+1}})\beta_{t-3}$ and $\mu_\gamma = (\omega_{0,CT_{t+1}}) + (\omega_{1,CT_{t+1}})\gamma_{t-1} + (\omega_{2,CT_{t+1}})\gamma_{t-2} + (\omega_{3,CT_{t+1}})\gamma_{t-3}$ are linear combinations of the three previous values of $\beta$ and $\gamma$, (respectively). The coefficients of those linear combinations depend on the value of the random variable $\mathrm{CT}_{t+1}$. We did not specify a distribution for the node `New_infections`$_{t+1}$ because its value is deterministically computed as $S_t - S_{t+1}$.

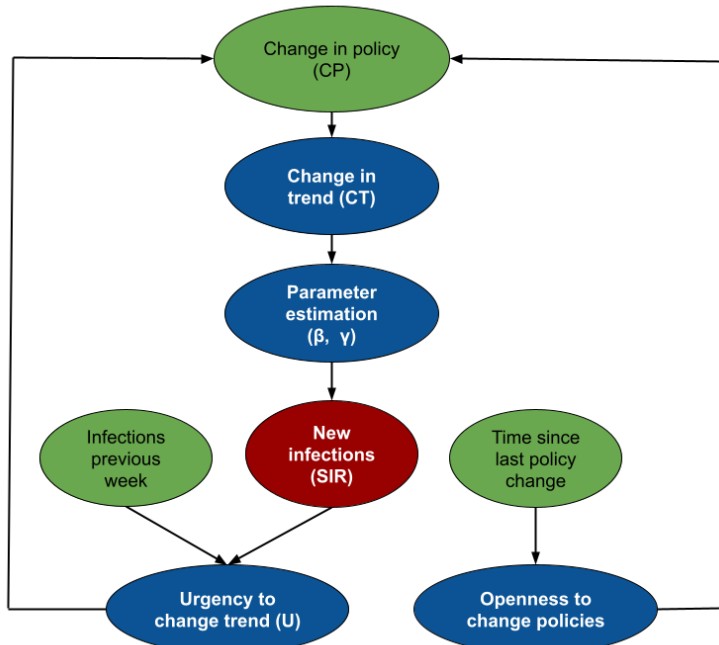

**Figure 2.** Intuition behind SIMLR. The policies currently in place determine the value of the parameters needed to infer the next values, using an SIR model. Those predictions are then used to estimate how the policies might change in the future.

Informally, the assignment $\mathrm{CT}_t = -1$ means that we expect a change in trend from an increasing number of infections to a decreasing one. The opposite happens when $\mathrm{CT}_t = 1$, while $\mathrm{CT}_t = 0$ means that we expect the population to follow the current trend (either increasing or decreasing). We assume these changes in trend depend on changes in the government policies 2 to 4 weeks prior to the week of our forecast—e.g., we use $\{\mathrm{CT}_{t-3}, \mathrm{CT}_{t-2}, \mathrm{CT}_{t-1}\}$ when predicting the number of new infections at $t + 1$, $\Delta I_{t+1}$, and we need $\{\mathrm{CT}_t, \mathrm{CT}_{t+1}, \mathrm{CT}_{t+2}\}$ when predicting $\Delta I_{t+4}$. Note that, at time $t$, we will not know $\mathrm{CT}_{t+1}$ nor $\mathrm{CT}_{t+2}$. We chose this interval based on the assumption that the incubation period of the virus is 2 weeks.

The status of $CT_{t+1}$ defines the coefficients that relate $\beta_{t+1}$ and $\gamma_{t+1}$ with their three previous values $\beta_t, \beta_{t-1}, \beta_{t-2}$ and $\gamma_t, \gamma_{t-1}, \gamma_{t-2}$, respectively. Since $\beta_{t+1}$ and $\gamma_{t+1}$ fully parameterize the SIR model in Equation (1), we can estimate the new number of infected people, $\Delta I_{t+1}$, from these parameters (as well as the SIR values at time $t$).

The random variables $U_t \in \{-1, 0, 1\}$ and $O_t \in \{0, 1\}$ are auxiliary variables designed to predict the probability of observing a change in policy at time $t + 1$. Intuitively, $U_t$ represents the "urgency" of modifying a policy. As the number of cases per 100K inhabitants and the rate of change between the number of cases in two consecutive time points increases, the urgency to set stricter government policies increases. As the number (and rate of change) of cases decreases, the urgency to relax the policies increases. Finally, $O_t$ models the "willingness" to execute a change in government policies. As the number of time points without a change increases, so does this "willingness".

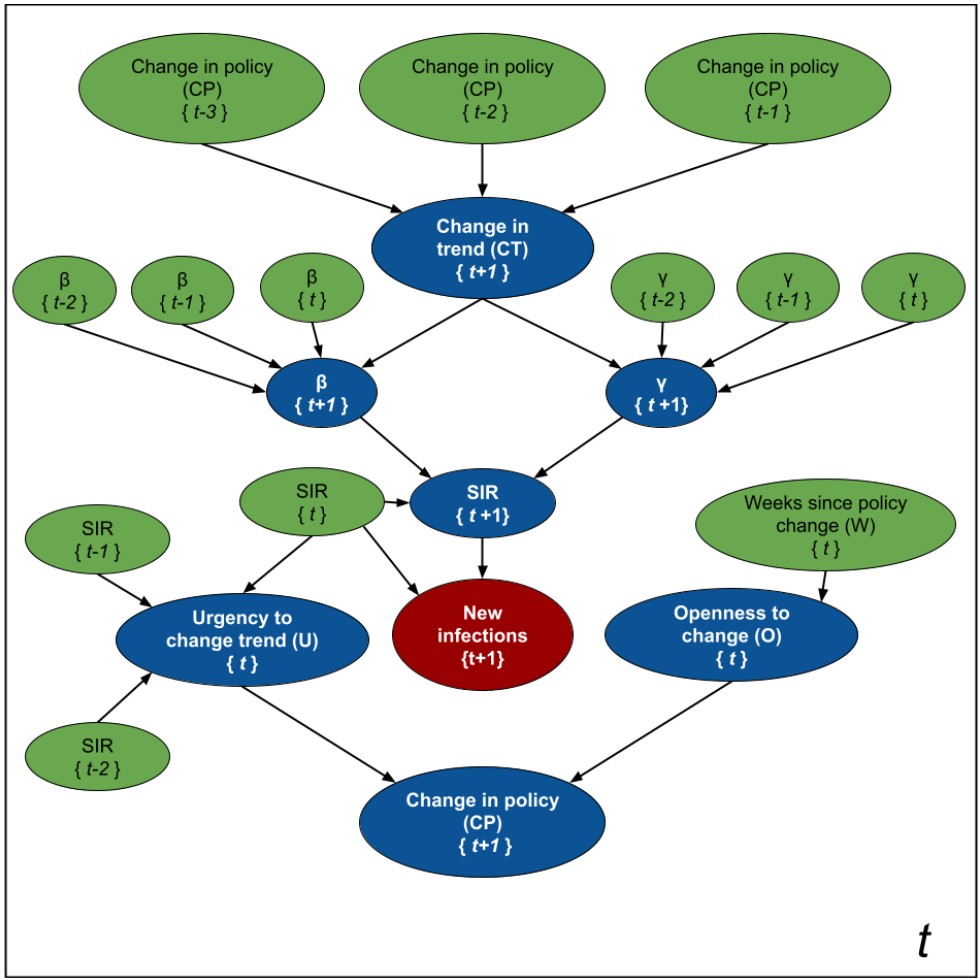

**Figure 3.** Modeling SIMLR as a PGM for forecasting new cases of COVID-19. The blue nodes are estimated at each time point, while the green ones are either based on past information, or where estimated in a previous iteration.

### 2.1. SIR with Time-Varying Parameters

We can approximate an SIR model by transforming the differential Equation (1) into the equations of differences:

$$
\begin{aligned}
S_t &= -\beta \frac{S_{t-1} I_{t-1}}{N} + S_{t-1} \\
I_t &= \beta \frac{S_{t-1} I_{t-1}}{N} - \gamma I_{t-1} + I_{t-1} \\
R_t &= \gamma I_{t-1} + R_{t-1}
\end{aligned}
\tag{3}
$$

where $S_t, I_t, R_t$ are the number of individuals in the groups Susceptible, Infected and Removed, respectively, at time $t$. Similarly $S_{t-1}, I_{t-1}, R_{t-1}$ represent the number individuals in each group at time $t-1$. $\beta$ is the transmission rate, and $\gamma$ is the recovery rate.

While the SIR model is non-linear with respect to the states (S, I, R), it is linear with respect to the parameters $\beta$ and $\gamma$. Therefore, under the assumption of constant and known population size (i.e., $N = S_t + I_t + R_t$) we can re-write the set of Equation (3) as:

$$
\begin{aligned}
\begin{bmatrix} S_t \\ I_t \end{bmatrix} &= \begin{bmatrix} -\frac{S_{t-1} I_{t-1}}{N} & 0 \\ \frac{S_{t-1} I_{t-1}}{N} & -I_{t-1} \end{bmatrix} \begin{bmatrix} \beta \\ \gamma \end{bmatrix} + \begin{bmatrix} S_{t-1} \\ I_{t-1} \end{bmatrix} \\
R_t &= N - S_t - I_t
\end{aligned}
\tag{4}
$$

Given a sequence of states $x_1, \ldots, x_n$, where $x_t = [S_t \ I_t]^T$, it is possible to estimate the optimal parameters of the SIR model as:

$$
(\beta^*, \gamma^*) = \underset{\beta, \gamma}{\arg\min} \sum_{i=1}^{n} ||x_i - \hat{x}_i||^2 + \lambda_1 (\beta - \beta_0)^2 + \lambda_2 (\gamma - \gamma_0)^2
\tag{5}
$$

where $\hat{x}_i$ is computed using Equation (4), and $\lambda_1$ and $\lambda_2$ are optional regularization parameters that allow the incorporation of the priors $\beta_0$ and $\gamma_0$. For the case of Gaussian priors—i.e., $\beta \sim \mathcal{N}(\beta_0, \sigma_\beta^2)$ and $\gamma \sim \mathcal{N}(\gamma_0, \sigma_\gamma^2)$—we use $\lambda_1 = \frac{1}{2\sigma_\beta^2}$ and $\lambda_2 = \frac{1}{2\sigma_\gamma^2}$ [28]. Intuitively, Equation (5) computes the transmission rate ($\beta^*$) and the recovery rate ($\gamma^*$) that best explain the number of new infections, deaths, and recovered people in a fixed time frame. If we know a standard recovery rate and transmission rate a priori ($\beta_0, \gamma_0$), it is possible to incorporate them into the Equation (5) as regularization parameters. The weights $\lambda_1$ and $\lambda_2$ control how much to weight those prior parameters. Small weights means we basically use the parameters learned by the data, and large weights mean more emphasis on the prior information.

In the traditional SIR model, we set $\lambda_1 = \lambda_2 = 0$ and fit a single $\beta$ and $\gamma$ to the entire time series. However, as shown in Figure 1a, an SIR model with fixed parameters is unable to accurately model several waves of infections. As illustration, Figure 1b shows the predictions produced by fitting an SIR with fixed parameters (Equation (5)) to the US data from 29 March 2020 to 3 May 2021, and then using those parameters to make predictions one week in advance, over this same interval. That is, using this learned ($\beta, \gamma$), and the number of people in the *S*, *I*, and *R* compartments on 28 March 2020, we predicted the number of observed cases during the week of 29 March 2020 to 4 April 2020. We repeated the same procedure for the entire time series. Note that even though the parameters $\beta$ and $\gamma$ were found using the entire time series – i.e., using information that was not available at the time of prediction—the resulting model still does a poor job fitting the reported data.

Figure 1c, on the other hand, was created by allowing $\beta$ and $\gamma$ to change every week. Here, we first found the parameters that fit the data from 29 March 2020 to 4 April 2020—call them $\beta_1$ and $\gamma_1$—then used those parameters along with the SIR state on 28 March 2020 to predict the number of new infections one week ahead—i.e., the sampled week of 29 March 2020 to 4 April 2020. By repeating this procedure during the entire time series we obtained an almost perfect fit to the data. Of course, these are also not "legal" predictions since they

too use information that is not available at prediction time—i.e., they used the number of reported infections during this first week to find the parameters, which were then used to estimate the number of cases over this time. However, this "cheating" example shows that an SIR model, with the optimal time-varying parameters, can model the complex dynamics of COVID-19. Recall from Figure 1b that this is not the case in the SIR model with fixed parameters, which cannot even properly fit the training data.

### 2.2. Estimating $\beta_{t+1}$ and $\gamma_{t+1}$

Naturally, the challenge is "legally" computing the appropriate values of $\beta_{t+1}$ and $\gamma_{t+1}$, for each week, using only the data that is known at time $t$. Figure 3 shows that computing $\beta_{t+1}$ and $\gamma_{t+1}$ depends on the status of the random variable $CT_{t+1}$. When $CT_{t+1} = 0$—i.e., there is no change in the current trend—we assume that:

$$\beta_{t+1} \sim \mathcal{N}(\alpha_0 + \alpha_1\beta_t + \alpha_2\beta_{t-1} + \alpha_3\beta_{t-2},\ \sigma_\beta^2)$$
$$\gamma_{t+1} \sim \mathcal{N}(\omega_0 + \omega_1\gamma_t + \omega_2\gamma_{t-1} + \omega_3\gamma_{t-2},\ \sigma_\gamma^2)$$

(6)

At time $t$, we can use the historical daily data $x_1, x_2, \ldots, x_t$ to find the weekly parameters $\beta_1, \beta_2, \ldots, \beta_{t/7}$ and $\gamma_1, \gamma_2, \ldots, \gamma_{t/7}$. Note that the is just one value for each week, so is there are 140 days, there are $140/7 = 20$ weeks. The first weekly pair $(\beta_1, \gamma_1)$ is found by fitting Equation (5) to $x_1, \ldots, x_7$; $(\beta_2, \gamma_2)$ to $x_8, \ldots, x_{14}$; and so on. Finally, we find the parameters $\boldsymbol{\alpha}$ and $\boldsymbol{\omega}$ in Equation (6) by maximizing the likelihood of the computed pairs. After finding those parameters, it is straightforward to infer $(\beta_{t+1}, \gamma_{t+1})$. Note that this approach is the probabilistic version of linear regression. To estimate the parameters $\sigma_\beta^2$ and $\sigma_\gamma^2$ we can simply estimate the variance of the residuals. An advantage of also computing these variances is that it is possible to obtain confidence intervals by sampling from the distribution in Equation (6) and then using those samples along with Equation (3) to estimate the distribution of the new infected people.

We estimated $\beta_{t+1}$ and $\gamma_{t+1}$ as a function of the 3 previous values of those parameters since this allows them to incorporate the velocity and acceleration at which the parameters change. We computed the velocity of $\beta$ as $v_{\beta,t} = \beta_t - \beta_{t-1}$ and its acceleration as $a_{\beta,t} = v_{\beta,t} - v_{\beta,t-1}$. Then, estimating $\beta_t = \theta_0 + \theta_1\beta_{t-1} + \theta_2 v_{\beta,t-1} + \theta_3 a_{\beta,t-1}$ is equivalent to the model in Equation (6). The same reasoning applies to the computation of $\gamma_t$. We call this approach the "trend-following varying-time parameters SIR", tf-v-SIR.

For the case of $CT_t = -1$ and $CT_t = 1$ (which represents a change in trend from increasing number of infections to decreasing number of infections or vice-versa), we set $\beta_{t+1}$ and $\gamma_{t+1}$ to values such that the predicted number of new cases at week $t + 1$ is identical to the one at week $t$. We call this the "Same as the Last Observed Week" (SLOW) model. As shown in Section 3, SLOW is a baseline with very good performance despite its simplicity. Given that the pandemic is a physical phenomenon that changes relatively slowly from one week to the next, making a prediction that assumes that the new number of cases will remain constant is not a bad prediction.

### 2.3. Estimating $CT_{t+1}$, $CP_{t+1}$, $O_t$

The random variables $CT_{t+1}, CP_{t+1}$ and $O_t$ in Figure 3 are all discrete nodes with discrete parents, meaning their probability mass functions are fully defined by conditional probability tables (CPTs). Learning the parameters of such CPTs from data is challenging due to the scarcity of historical information. The random variable $CT_{t+1}$ depends on the random variable changes in policy (CP) at times $t - 1, t - 2, t - 3$; however, there are very few changes in policy in a given region, meaning it is difficult to accurately estimate those probabilities from data. For the random variable O, which represents the "willingness" of the government to implement a change in policy, there is no observable data at all. We therefore relied on prior expert knowledge to set the parameters of the conditional probability tables for these random variables. Figure 4 shows the conditional probability

tables (CPT) for the random variables $CT_{t+1}$, $CP_{t+1}$, $O_t$. The intuition used to generate the CPT's is as follows:

We considered that a change in trend in the current week depends on changes in policies during the previous three weeks. We chose 3 weeks using the hypothesis that the incubation period for the virus is 2 weeks. Then the effects of a policy will be reflected approximately 2 weeks after a change. We decided to analyze also one week after, and one week before this period, giving as a result the tracking of $CP_{t-3}$ to $CP_{t-1}$. Secondly, we also assume that whenever we observe a change of policy that will move the trend from going up to going down, then that event will most likely happen. This is why most of the probability mass is located in a single column. For example, if we observe that the policies are relaxed at any point during the weeks $t - 3$, $t - 2$, or $t - 1$, then we assume that we will observe a change in trend with 99.9% probability.

The rationale for the CPT $P(O_t \mid W_t)$ is that the government becomes more open to implement changes after long periods of 'inactivity'. For example, if they implement a change in policy this week ($W_t = 0$), then the probability of considering a second change of policy during the same week is very small (0.01%). We are assuming that, after a change in policy, the government will wait to see the effect of that change before taking further action. If 4 weeks have passed since the last change in policy, we estimated the probability of considering a change in the policy as 50%, while if more than 7 weeks have passed, then they are fully open to the possibility of implementing a new change.

$P(O_t \mid W_t)$ estimates the probability of considering a change in the policy. The probability of actually implementing a change, $P(CP_{t+1} \mid O_t, U_t)$ depends not only on how willing the government is, but also on how urgent it is to make a change. In general, if the government is open to implement a change, and the urgency is "high", then the probability of changing a policy is high. We also considered that the government "prefers" to either not make changes in policy or relax the policies, rather than to implement more strict policies.

| CP t-3 | CP t-2 | CP t-1 | $P(CT_{t+1} = x \mid CP_{t-3}, CP_{t-2}, CP_{t-1})$ | | |
|---|---|---|---|---|---|
| | | | x = -1 | x = 0 | x = 1 |
| -1 | -1 | -1 | 0.999 | 0.0005 | 0.0005 |
| -1 | -1 | 0 | 0.999 | 0.0005 | 0.0005 |
| -1 | -1 | 1 | 0.999 | 0.0005 | 0.0005 |
| -1 | 0 | -1 | 0.999 | 0.0005 | 0.0005 |
| -1 | 0 | 0 | 0.999 | 0.0005 | 0.0005 |
| -1 | 0 | 1 | 0.0005 | 0.0005 | 0.999 |
| -1 | 1 | -1 | 0.999 | 0.0005 | 0.0005 |
| -1 | 1 | 0 | 0.0005 | 0.0005 | 0.999 |
| -1 | 1 | 1 | 0.0005 | 0.0005 | 0.999 |
| 0 | -1 | -1 | 0.999 | 0.0005 | 0.0005 |
| 0 | -1 | 0 | 0.999 | 0.0005 | 0.0005 |
| 0 | -1 | 1 | 0.0005 | 0.0005 | 0.999 |
| 0 | 0 | -1 | 0.999 | 0.0005 | 0.0005 |
| 0 | 0 | 0 | 0.0005 | 0.999 | 0.0005 |
| 0 | 0 | 1 | 0.0005 | 0.0005 | 0.999 |
| 0 | 1 | -1 | 0.999 | 0.0005 | 0.0005 |
| 0 | 1 | 0 | 0.0005 | 0.0005 | 0.999 |
| 0 | 1 | 1 | 0.0005 | 0.0005 | 0.999 |
| 1 | -1 | -1 | 0.999 | 0.0005 | 0.0005 |
| 1 | -1 | 0 | 0.999 | 0.0005 | 0.0005 |
| 1 | -1 | 1 | 0.0005 | 0.0005 | 0.999 |
| 1 | 0 | -1 | 0.999 | 0.0005 | 0.0005 |
| 1 | 0 | 0 | 0.0005 | 0.0005 | 0.999 |
| 1 | 0 | 1 | 0.0005 | 0.0005 | 0.999 |
| 1 | 1 | -1 | 0.0005 | 0.0005 | 0.999 |
| 1 | 1 | 0 | 0.0005 | 0.0005 | 0.999 |
| 1 | 1 | 1 | 0.0005 | 0.0005 | 0.999 |

| W t | $P(O_t = x \mid W_t)$ | |
|---|---|---|
| | x = 0 | x = 1 |
| 0 | 0.9999 | 0.0001 |
| 1 | 0.9 | 0.1 |
| 2 | 0.85 | 0.15 |
| 3 | 0.75 | 0.25 |
| 4 | 0.5 | 0.5 |
| 5 | 0.25 | 0.75 |
| 6 | 0.0001 | 0.9999 |
| 7+ | 0.0001 | 0.9999 |

| O t | U t | $P(CP_{t+1} = x \mid O_t, U_t)$ | | |
|---|---|---|---|---|
| | | x = -1 | x = 0 | x = 1 |
| 0 | -1 | 0.02 | 0.97 | 0.01 |
| 0 | 0 | 0.005 | 0.99 | 0.005 |
| 0 | 1 | 0.01 | 0.97 | 0.02 |
| 1 | -1 | 0.8 | 0.19 | 0.01 |
| 1 | 0 | 0.09 | 0.9 | 0.01 |
| 1 | 1 | 0.01 | 0.24 | 0.75 |

**Figure 4.** Conditional probability tables used by SIMLR. The names of the variables refer to the nodes that appear on Figure 2 on the main text.

## 2.4. Estimating $U_t$

For modelling the random variable $U_t$, which represents the "Urgency to change the trend", we use an NN-CPD (neural-network conditional probability distribution), which is a modified version of the multinomial logistic conditional probability distribution [29].

**Definition 1** (NN-CPD). *Let* $Y \in \{1, \ldots, m\}$ *be an m-valued random variable with* k *parents* $X_1, \ldots, X_k$ *that each take on numerical values. The conditional probability distribution* $P(Y \mid X_1, \ldots, X_k)$ *is an NN-CPD if there is an function* $z = f_\theta(X_1, \ldots, X_k) \in \mathbb{R}^m$, *represented as a neural network with parameters* $\theta$, *such that* $p(Y = i \mid x_1, \ldots, x_k) = \exp(z_i) / \sum_j \exp(z_j)$, *where* $z_i$ *represents the i-th entry of* $z$.

Note $U_t$ is a latent variable, so there is no observable data at all. We again rely on domain knowledge to estimate its probabilities. To compute $P(U_t \mid \mathrm{SIR}_{t-2}, \mathrm{SIR}_{t-1}, \mathrm{SIR}_t)$, we extract two features: $c_t = 10 \times 10^5 (S_{t-1} - S_t)/N$, which represents the number of new reported infections per 100K inhabitants; and $v_t = c_t - c_{t-1}$, which estimates the rate of change of $c_t$. Then define $P(U_t \mid \mathrm{SIR}_{t-2}, \mathrm{SIR}_{t-1}, \mathrm{SIR}_t) = P(U_t \mid c_t, v_t)$.

To learn the parameters $\theta$ we created the dataset shown in Figure 5. Note that the targets in such dataset are probabilities. We relied on the probabilistic labels approach proposed by Vega et al. [30] to use a dataset with few training instances along with their probabilities to learn the parameters of a neural network more efficiently. We trained and a simple neural network with a single hidden layers with 64 units, and 3 output units with softmax activation.

The random variables $U_t \in \{-1, 0, 1\}$ and $O_t \in \{0, 1\}$ are auxiliary variables designed to predict the probability of observing a change in policy at time $t + 1$. Intuitively, $U_t$ represents the "urgency" of modifying a policy. As the number of cases per 100 K inhabitants and the rate of change between the number of cases in two consecutive time points increases, the urgency to set stricter government policies increases. As the number (and rate of change) of cases decreases, the urgency to relax the policies increases. Most of the parameters in both NN-CPD tables are similar for the US and Canada, the difference arises from a perceived preference for not setting very strict policies in the US during the first year of the pandemic.

NN- CPD for US

| $C_t$ | $V_t$ | P (U t = x \| C t, V t) | | |
|---|---|---|---|---|
| | | x = -1 | x = 0 | x = 1 |
| 0 | -40 | 1 | 0 | 0 |
| 50 | -20 | 0.9 | 0.1 | 0 |
| 50 | 0 | 0.05 | 0.95 | 0 |
| 50 | 20 | 0 | 1 | 0 |
| 125 | -20 | 0.05 | 0.95 | 0 |
| 125 | 0 | 0.01 | 0.98 | 0.01 |
| 125 | 20 | 0 | 0.95 | 0.05 |
| 200 | -20 | 0 | 1 | 0 |
| 200 | 0 | 0 | 0.95 | 0.05 |
| 200 | 20 | 0 | 0.1 | 0.9 |
| 200 | 40 | 0 | 0 | 1 |
| 250 | 40 | 0 | 0 | 1 |

NN- CPD for Canada (All provinces)

| $C_t$ | $V_t$ | P (U t = x \| C t, V t) | | |
|---|---|---|---|---|
| | | x = -1 | x = 0 | x = 1 |
| 0 | -40 | 1 | 0 | 0 |
| 50 | -20 | 0.9 | 0.1 | 0 |
| 50 | 0 | 0.5 | 0.5 | 0 |
| 50 | 20 | 0 | 1 | 0 |
| 125 | -20 | 0.5 | 0.5 | 0 |
| 125 | 0 | 0.01 | 0.98 | 0.01 |
| 125 | 20 | 0 | 0.5 | 0.5 |
| 200 | -20 | 0 | 1 | 0 |
| 200 | 0 | 0 | 0.5 | 0.5 |
| 200 | 20 | 0 | 0.1 | 0.9 |
| 200 | 40 | 0 | 0 | 1 |
| 250 | 40 | 0 | 0 | 1 |

**Figure 5.** Dataset used to create the NN-CPD for the variable $U_t$ and its visualization. Values closer to 1 (yellow) increase $p(U_t = 1 \mid C_t, V_t)$. Values closer to 0 (green) increase $p(U_t = 0 \mid C_t, V_t)$. Values closer to $-1$ (blue) increase $p(U_t = -1 \mid C_t, V_t)$.

*2.5. Evaluation*

We evaluated the performance of SIMLR, in terms of the mean absolute percentage error (MAPE) and mean absolute error (MAE), for forecasting the number of new infections one to four weeks in advance, in data from United States (as a country and individually for every state) and the six biggest provinces of Canada: Alberta (AB), British Columbia (BC), Manitoba (MN), Ontario (ON), Quebec (QB), and Saskatchewan (SK). For each of the regions, the predictions are done on a weekly basis, over the 39 weeks from 26 July 2020 to 1 May 2021. This time span captures different waves of infections. Equation (7) show the computation of the metrics used for evaluating our approach.

$$
\text{MAPE} = \frac{1}{n} \sum_{t=1}^{n} \left| \frac{y_t - \hat{y}_t}{y_t} \right|
$$
$$
\text{MAE} = \frac{1}{n} \sum_{t=1}^{n} |y_t - \hat{y}_t|
$$
(7)

At the end of every week, we fitted the SIMLR parameters using the data that was available until that week. For example, on 25 July 2020, we used all the data available from 1 January 2020 to 25 July 2020 to fit the parameters of SIMLR. Then, we made the predictions for the number of new infections during the weeks: 26 July 2020–1 August 2020 (one week in advance), 2 August 2020–8 August 2020 (two weeks in advance), 9 August 2020–15 August 2020 (three weeks in advance), and 16 August 2020–22 August 2020 (four weeks in advance). After this, we then fitted the parameters with data up to 1 August 2020 and repeated the same process, for 38 more iterations, until we covered the entire range of predictions.

We compared the performance of SIMLR with the SIR compartmental model with time-varying parameters learned using Equation (6) but no other random variable (tf-v-SIR), and with the simple model that forecasts that the number of cases one to four weeks in advance is the "Same as the Last Observed Week" (SLOW). For the United States data, we also compared the performance of SIMLR against the publicly available predictions at the COVID-19 Forecast Hub, which are the predictions submitted to the Center for Disease Control and Prevention (CDC) [31].

For training, we used the publicly available dataset OxCGRT [4], which contains the policies implemented by different regions, as well as the time period over which they were implemented. We limited our analysis to three policy decisions: `Workplace closing`, `Stay at home requirements`, and `Cancellation of public events` in the case of Canada. For the case of the United States we used `Restrictions on gatherings`, `Vaccination policy`, and `Cancellation of public events`. For information about the new number of reported cases and deaths, we used the publicly available COVID-19 Data Repository by the Center for Systems Science and Engineering at Johns Hopkins University [1]. The code for reproducing the results presented here are discussed in Appendix A.

## 3. Results

*3.1. Data Preprocessing*

Before inputting the time-series data to SIMLR, we performed some basic preprocessing during the training phase, and exclusively on the training data. We evaluated of our models by comparing its predictions with the results reported by the different health agencies –i.e., we did not fill in the data on the test sets:

1.  The original data contains the cumulative number of reported infections/deaths on a daily basis. We trivially transformed this time-series into the number of new daily infections/deaths.

2.   We considered negative values from the new daily infections/deaths time-series as missing, assuming these negative values arose due to inconsistencies during the data reporting procedure.
3.   We "filled-in" the missing values. When the number of new infections/deaths was missing at day $d$, we assumed that the entry at $d + 1$ contained the cases for both $d$ and $d + 1$, and divided the number of new infections/deaths evenly between both days.
4.   We eliminated outliers. For each day $d$, with number of reported new infections, $\Delta I_d$, we computed the mean ($\mu_d$) and standard deviation ($\sigma_d$ of the set $\Delta I_{d-10}, \ldots, \Delta I_{d-1}$; we then set $\Delta I_d := \min\{\Delta I_d, \mu_d + 4\sigma_d\}$.
5.   We used the number of new infections and new deaths to produce the SIR vector $SIR_t = [S_t, I_t, R_t]$.

In step 5, we assumed that everyone in a given region was susceptible at the start time—i.e., $S_0 = N$. At each new time point, we transfer the number of new infections from $S$ to $I$, and the number of new deaths and recovered from $I$ to $R$. If the number of new recovered people is not reported, we used the surveillance definition of recovered used by Canadian health agencies. This definition is based on the assumption that a recovered person is one who is not hospitalized and is 14 days past the day when they tested positive [32,33]:

> "Active and recovered status is a surveillance definition to try to understand the number of active cases in the population. It is not related to clinical management of cases. It is based on the assumption that a case is recovered 14 days after a particular date..."

### 3.2. MAPE and MAE

Figure 6 shows the MAPE of the one- to four-week forecasts for the United States as a country and the six biggest provinces of Canada. Note that SIMLR has a consistently lower MAPE than tf-v-SIR and SLOW. Figure 7 shows a similar result in terms of MAE. Tables 1 and 2 show the mean and standard deviations of the metrics corresponding to the Figures 6 and 7. In addition Table 3 show the correlation coefficient between the time series of the reported new infections every week and the predictions made by the different models.

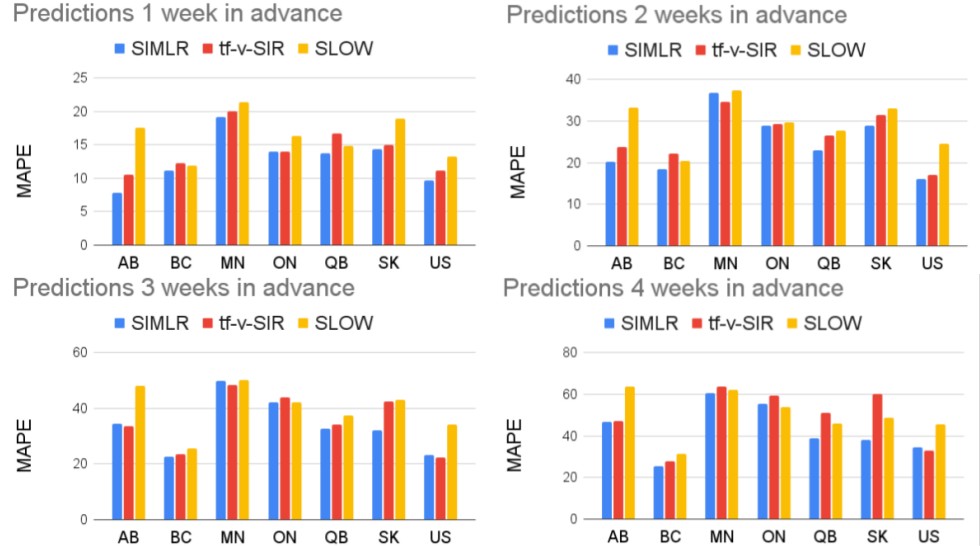

**Figure 6.** Comparison of SIMLR, SIR model with time-varying parameters, and SLOW. Table 1 contains the numerical information.

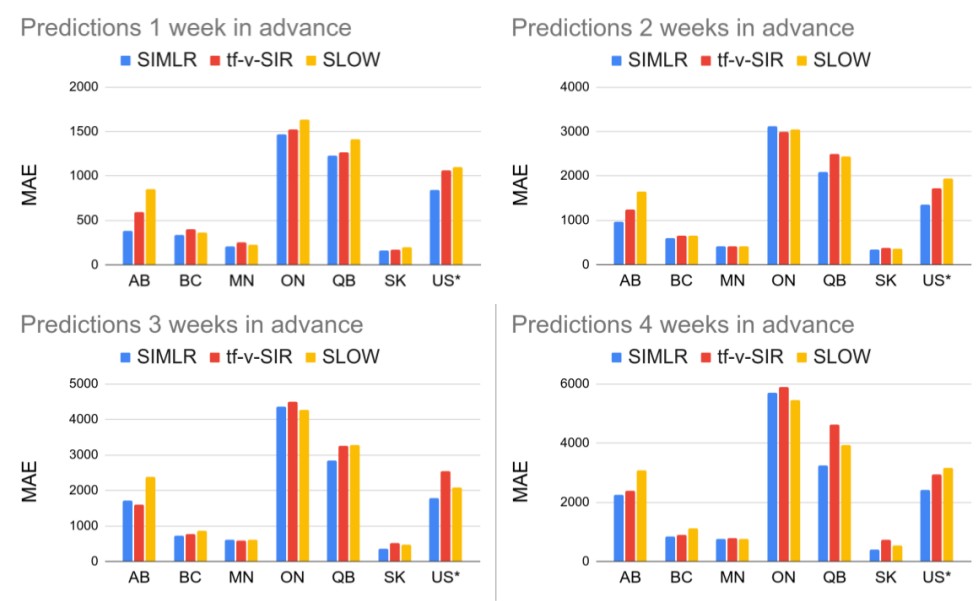

**Figure 7.** Comparison of SIMLR, SIR model with time-varying parameters, and SLOW in terms of MAE. To make the numbers comparable, the figures each show the US MAE values divided by 100.

**Table 1.** MAPE of the six biggest provinces in Canada and United States as a country, one- to four-weeks in advance. The number in parenthesis is the standard deviation.

|  | **Week 1** | | | **Week 2** | | |
|---|---|---|---|---|---|---|
|  | **SIMLR** | **tf-v-SIR** | **SLOW** | **SIMLR** | **tf-v-SIR** | **SLOW** |
| AB | 7 (8) | 10 (10) | 17 (9) | 20 (14) | 23 (16) | 33 (17) |
| BC | 11 (8) | 12 (10) | 11 (8) | 18 (10) | 22 (15) | 20 (13) |
| MN | 19 (14) | 20 (13) | 21 (15) | 36 (24) | 34 (22) | 37 (24) |
| ON | 14 (9) | 14 (10) | 16 (10) | 28 (21) | 29 (24) | 29 (19) |
| QB | 13 (11) | 14 (11) | 16 (11) | 23 (20) | 26 (30) | 27 (19) |
| SK | 14 (9) | 15 (12) | 18 (13) | 28 (17) | 31 (18) | 33 (18) |
| US | 9 (6) | 11 (8) | 13 (9) | 16 (13) | 19 (16) | 24 (17) |
|  | **Week 3** | | | **Week 4** | | |
|  | **SIMLR** | **tf-v-SIR** | **SLOW** | **SIMLR** | **tf-v-SIR** | **SLOW** |
| AB | 34 (21) | 33 (22) | 48 (26) | 46 (35) | 47 (33) | 63 (35) |
| BC | 22 (14) | 23 (16) | 25 (18) | 25 (20) | 27 (21) | 31 (20) |
| MN | 49 (31) | 48 (34) | 50 (27) | 60 (38) | 63 (42) | 62 (33) |
| ON | 42 (37) | 44 (40) | 42 (30) | 55 (51) | 59 (58) | 53 (40) |
| QB | 32 (28) | 34 (36) | 37 (27) | 38 (41) | 51 (64) | 45 (35) |
| SK | 32 (23) | 42 (32) | 43 (22) | 38 (24) | 60 (50) | 49 (26) |
| US | 23 (23) | 25 (26) | 34 (28) | 36 (38) | 38 (41) | 45 (40) |

**Table 2.** MAE of the six biggest provinces in Canada and United States as a country, one- to four-weeks in advance. The number in parenthesis is the standard deviation. For the case of the US the number of cases was divided by 100.

|  | Week 1 | | | Week 2 | | |
|---|---|---|---|---|---|---|
|  | **SIMLR** | **tf-v-SIR** | **SLOW** | **SIMLR** | **tf-v-SIR** | **SLOW** |
| AB | 385 (559) | 598 (905) | 850 (724) | 966 (971) | 1245 (1430) | 1651 (1258) |
| BC | 339 (304) | 397 (426) | 361 (294) | 594 (443) | 661 (480) | 648 (485) |
| MN | 204 (227) | 252 (271) | 221 (224) | 422 (371) | 418 (379) | 413 (346) |
| ON | 1471 (1343) | 1520 (1662) | 1635 (1388) | 3124 (2632) | 3001 (2847) | 3044 (2351) |
| QB | 1229 (1443) | 1265 (1354) | 1410 (975) | 2098 (2264) | 2496 (3270) | 2446 (1743) |
| SK | 161 (161) | 171 (203) | 194 (174) | 339 (294) | 382 (324) | 355 (264) |
| US* | 841 (796) | 1061 (1149) | 1103 (913) | 1361 (1398) | 1729 (1979) | 1933 (1580) |
|  | Week 3 | | | Week 4 | | |
|  | **SIMLR** | **tf-v-SIR** | **SLOW** | **SIMLR** | **tf-v-SIR** | **SLOW** |
| AB | 1719 (1381) | 1601 (1558) | 2378 (1649) | 2261 (1863) | 2385 (2087) | 3074 (1858) |
| BC | 731 (566) | 777 (672) | 853 (716) | 835 (709) | 883 (703) | 1127 (892) |
| MN | 609 (504) | 591 (501) | 602 (467) | 749 (612) | 775 (630) | 753 (571) |
| ON | 4357 (3672) | 4511 (3983) | 4266 (3053) | 5702 (4427) | 5910 (4988) | 5447 (3417) |
| QB | 2854 (2527) | 3261 (4096) | 3288 (2389) | 3244 (3131) | 4636 (6115) | 3947 (2788) |
| SK | 351 (320) | 522 (500) | 472 (306) | 410 (287) | 736 (733) | 541 (348) |
| US* | 1793 (2012) | 2089 (2768) | 2538 (2151) | 2414 (2755) | 2933 (4027) | 3157 (2679) |

**Table 3.** Pearson correlation coefficient between the ground truth and the predictions of the six biggest provinces in Canada and United States as a country one- to four-weeks in advance.

|  | Week 1 | | | Week 2 | | |
|---|---|---|---|---|---|---|
|  | **SIMLR** | **tf-v-SIR** | **SLOW** | **SIMLR** | **tf-v-SIR** | **SLOW** |
| AB | 0.99 | 0.98 | 0.96 | 0.94 | 0.95 | 0.84 |
| BC | 0.97 | 0.97 | 0.97 | 0.90 | 0.89 | 0.90 |
| MN | 0.96 | 0.96 | 0.95 | 0.85 | 0.87 | 0.86 |
| ON | 0.96 | 0.97 | 0.96 | 0.83 | 0.85 | 0.85 |
| QB | 0.97 | 0.97 | 0.96 | 0.93 | 0.89 | 0.86 |
| SK | 0.97 | 0.96 | 0.95 | 0.89 | 0.89 | 0.86 |
| US | 0.97 | 0.97 | 0.96 | 0.93 | 0.93 | 0.87 |
|  | Week 3 | | | Week 4 | | |
|  | **SIMLR** | **tf-v-SIR** | **SLOW** | **SIMLR** | **tf-v-SIR** | **SLOW** |
| AB | 0.90 | 0.90 | 0.68 | 0.84 | 0.85 | 0.50 |
| BC | 0.84 | 0.83 | 0.83 | 0.80 | 0.81 | 0.75 |
| MN | 0.69 | 0.75 | 0.73 | 0.51 | 0.56 | 0.57 |
| ON | 0.67 | 0.68 | 0.71 | 0.51 | 0.53 | 0.58 |
| QB | 0.83 | 0.84 | 0.74 | 0.71 | 0.71 | 0.61 |
| SK | 0.82 | 0.81 | 0.78 | 0.73 | 0.69 | 0.71 |
| US | 0.88 | 0.90 | 0.77 | 0.80 | 0.84 | 0.65 |

Figure 8c shows how our proposed SIMLR approach compares with the 18 models that submitted predictions at the country level to the CDC during the same span of time (results at the state level are included in the Appendix B). Note that SIMLR and the model *LNQ-ens1* are the best performing models, with no statistically significant difference ($p > 0.05$ on a paired *t*-test) with respect to MAPE.

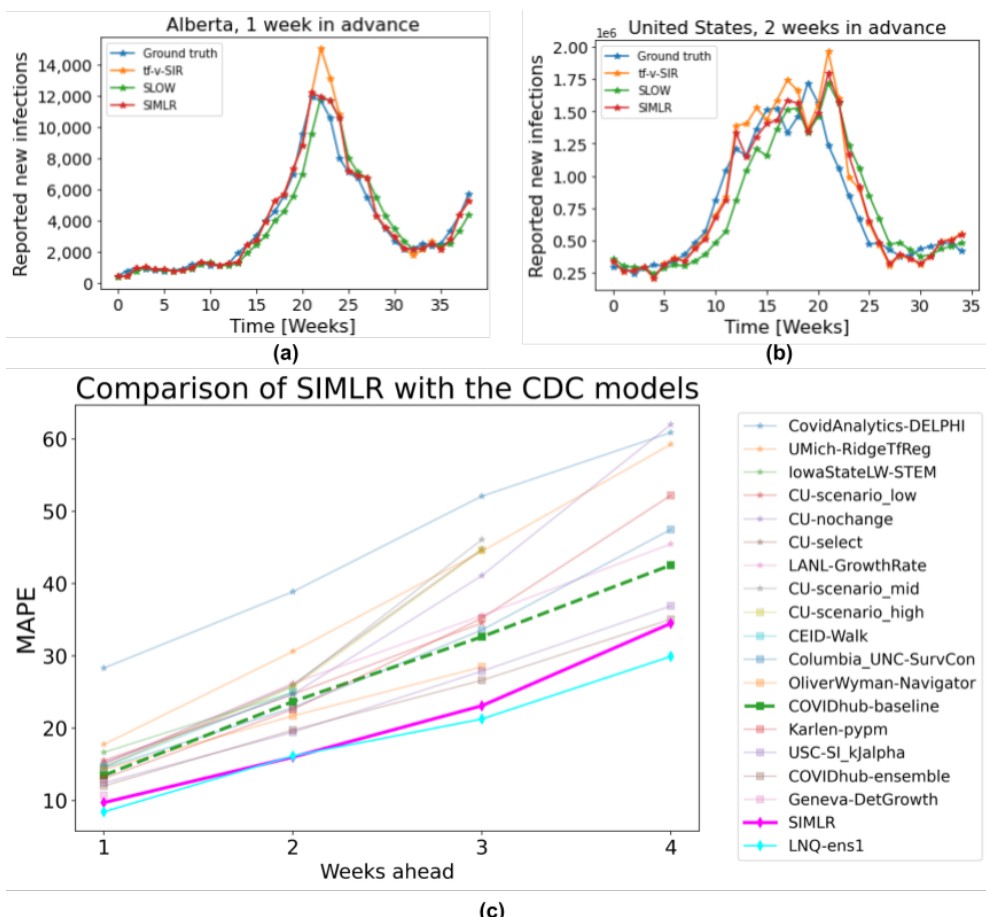

**Figure 8.** (**a**) 1-week forecasts SIMLR, tf-v-SIR, and SLOW, for Alberta, Canada. (**b**) 2-week forecasts, of the same models, for US data. (**c**) Comparison of SIMLR versus models submitted to the CDC (on US data).

## 4. Discussion

Figure 8 illustrates the actual predictions of SIMLR one week in advance for the province of Alberta, Canada; and two weeks in advance for the US as a country. These two cases exemplify the behaviour of SIMLR. As noted above, there is a 2- to 4-week lag after a policy changes, before we see the effects. This means the task of making 1-week forecasts is relatively simple, as the relevant policy (at times $t - 3$ to $t - 1$) is fully observable. This allows SIMLR to directly compute $CT_{t+1}$, which can then choose whether to continue using the SIR with time-varying parameters if no policy changed at time $t - 1$, $t - 2$, or $t - 3$, or using the SLOW predictor if the policy changed.

Figure 8a shows a change in the trend of reported new cases at week 22. However, just by looking at the evolution of number of new infections before week 22, there is no way to predict this change, which is why tf-v-SIR predicts that the number of new infections will continue growing. However, since SIMLR observed a change in the government policies at week 20, it realized it could no longer rely on its estimation of parameters and so switched to the SLOW model, which is why it was more accurate here. A similar behaviour occurs in week 34, when the third wave of cases in Alberta started. Due to a relaxation in the policies on week 31, SIMLR (at week 31) correctly predicted a change of trend around weeks 33–35.

This behavior is not exclusive for the data of Alberta and it explains why the performance of SIMLR is consistently higher than the baselines used for comparison in Figure 6 and Figure 8c. A striking result is how hard it is to beat the simple SLOW model (COVIDhub-baseline). Out of the 19 models considered here, only five (including SIMLR) do better than this simple baseline when predicting three to four weeks ahead. This brings some insight into the challenge of making accurate prediction in the medium term—probably due to the need to predict, then use, policy change information. Tables A1–A4 in the Appendix B show a comparison between our proposed SIMLR and tf-v-SIR against the models submitted to the CDC for all the states in the US. SIMLR consistently ranks among the best performers, with the advantage of being an interpretable model.

A deeper analysis of Tables A1–A4 shows that, in some states, the performance of SIMLR degrades for longer range predictions. This occurs because we are monitoring only the same three policies for all the states; however, different states might have implemented different policies and reacted differently to them. For example, closing schools might be a relevant policy in a state where there is an outbreak that involves children, but not as relevant if most of the cases are in older people.

Tracking irrelevant policies might degrade the performance of SIMLR. If the status of an irrelevant policy changes, then the dynamics of the disease will not be affected. The model however, will assume that the change in the policy will cause a change of trend and it will rely on the SLOW model, instead of the more accurate tf-v-SIR. Although SIMLR can be adapted to track different policies, the policies that are relevant for a given state must be given as an input. So while we think our overall approach applies in general, our specific model (tracking these specific policies, etc.) might not perform accurate predictions across all the regions. This is also a strength, in that it is trivial to adapt our specific model to track the policies of interest within a given region.

Predictions at the country level are more complicated, since most of the time policies are implemented at the state (or province) level instead of nationally. For making predictions for an entire country, as well as predictions three or four weeks in advance, SIMLR first predicts, then uses, the likelihood of observing a change in trend, at every week. In these cases, the random variable $CT_{t+1}$ no longer acts like a "switch", but instead it mixes the predictions of the tf-v-SIR and SLOW models, according to the probability of observing a change in the trend.

Figure 8b shows that whenever there is a stable trend in the number of new reported infections—which suggests there have been no recent policy changes—SIMLR relies on the predictions of the tf-v-SIR model; however, as the number (and rate of change) of new infections increases, so does the probability of observing a change in the policy. Therefore, SIMLR starts giving more weight to the predictions of the SLOW model. Note this behavior in the same figure during weeks 13–20.

One limitation of SIMLR is that it relies on conditional probabilities that are hard to learn due to lack of data, which forced us to build them based on domain knowledge. If this prior knowledge is inaccurate, then the predictions might be also misleading. Also, different regions might have different "thresholds" for taking action. Despite this limitation, SIMLR produced state-of-the-art results in both forecasting in the US as a country and at the provincial level in Canada, as well as very competitive results in predictions at the state level in the US.

Note that modelling SIMLR as a PGM does not imply causality. Although changes in the observed policy influence changes in the trend of new reported cases, the opposite is also true in reality. However, using probabilistic graphical models does makes it interpretable. It also allows us to incorporate domain knowledge that compensates for the relatively scarce data. SIMLR's excellent performance—comparable to state-of-the-art systems in this competitive task—show that it is possible to design interpretable machine learning models without sacrificing performance.

## 5. Conclusions

Forecasting the number of new COVID-19 infections is a very challenging task. Many factors play a role on how the disease spreads, including the government policies and the adherence of citizens to such policies. These elements are difficult to model mathematically; however, the collected data (number of new infections and deaths, for example) are a reflection of all those complex interactions.

Machine learning, on the other side, excels at learning patterns directly from the data. Unfortunately, training many models from scratch can require a great deal of data, especially to learn complex patterns, such as the evolution of a pandemic.

We proposed SIMLR, a methodology that uses machine learning (ML) techniques to learn a model that can set, and adjust, the parameters of mathematical model for epidemiology (SIR). SIMLR augments that SIR model by incorporating expert knowledge in the form of a probabilistic graphical model. In this way, human experts can incorporate their believes in the likelihood that a policy will change, and when. By combining both components we substantially reduce the data that machine learning usually requires to produce models that can make accurate predictions.

Importantly, besides providing state-of-the-art predictions in terms of MAPE in the short and medium term, the resulting SIMLR model is interpretable and probabilistic. The first means that we can justify the predictions given by the algorithm—e.g., "SIMLR predicts 1000 cases for the next week due to a change in the government policies that will decrease the transmission rate". The second means we can produce probabilistic values—so instead of predicting a single value, it can predict the entire probability distribution—e.g., the probability of 100 cases next week, or of 200 cases or of 1000, etc.

This paper demonstrated that a model that explicitly models and incorporates government policy decisions can accurately produce one- to four-week forecasts of the number of COVID-19 infections. This involved showing that an SIR model with time-varying parameters is enough to describe the complex dynamics of this pandemic, including the different waves of infections. We expect that this approach will be useful not only for modelling COVID-19, but other infectious diseases as well. We also hope that its interpretability will leads to its adoption by researchers, and users, in epidemiology and other non-ML fields.

**Author Contributions:** Conceptualization, R.V., L.F. and R.G.; methodology, R.V., L.F. and R.G.; software, R.V.; validation, R.V.; formal analysis, R.V., L.F. and R.G.; investigation, R.V., L.F. and R.G.; resources, R.G.; data curation, R.V.; writing—original draft preparation, R.V.; writing—review and editing, R.V., L.F. and R.G.; visualization, R.V.; supervision, R.G.; project administration, R.G.; funding acquisition, R.G. All authors have read and agreed to the published version of the manuscript.

**Funding:** This research was funded by Alberta Machine Intelligence Institute.

**Institutional Review Board Statement:** Not applicable.

**Informed Consent Statement:** Not applicable.

**Data Availability Statement:** All the datasets used for this manuscript are publicly available. For information about the new number of reported cases and deaths, we used the publicly available COVID-19 Data Repository by the Center for Systems Science and Engineering at Johns Hopkins University [1] https://github.com/CSSEGISandData/COVID-19, accessed on 1 September 2020. For policy tracking we used the OxCGRT [4] https://github.com/OxCGRT/covid-policy-tracker, accessed on 1 September 2020. For comparing our approach with other models we used the publicly available predictions at the COVID-19 Forecast Hub [31] https://github.com/reichlab/covid19-forecast-hub, accessed on 1 September 2020.

**Acknowledgments:** We thank the Google Cloud Research Credits program and Compute Canada for providing computational support. We also benefited from our many meetings with our colleagues of the greater University of Alberta Covid-Team.

**Conflicts of Interest:** The authors declare no conflict of interest.

## Appendix A. Code Availability

The code for reproducing the main results of this manuscript are publicly available at: https://github.com/rvegaml/SIMLR, accessed on 7 December 2021.

There are six jupyter notebooks on that repository. All the experiments were run using an e2-standard-4 (4 vCPUs, 16 GB memory) computer in the Google Cloud Platform.

- CDC_models.ipynb: It contains the code used to compile the predictions of the models submitted to the CDC. The dataset required to run this script was not included due to the size, but it is publicly available.
- Comparison_CDC.ipynb: It contains the code to create the graphs that compare SIMLR with the models submitted to the CDC. It uses the files created by the previous notebook.
- Model_Canada_Provinces.ipynb: It contains the data to predict the number of cases 1 to 4 weeks in advance in the 6 biggest provinces in Canada.
- Model_US_Country.ipynb: Similar to the previous one, but for the predictions on US at the country level.
- Model_US_States.ipynb: Similar to the previous one, but for the predictions on US at the state level.
- SIR_Simulations.ipynb: Code to create the simulated SIR, and to show how a simple SIR model with time-varying parameters can describe the complexities of the COVID-19 dynamics.

The provided repository in addition contains the in-house developed python library *MLib*. This library contains custom code for inference in probabilistic graphical models.

## Appendix B. Additional Tables

**Table A1.** Comparison of MAPE between different models across all the states in the US 1 week in advance. The number in parenthesis represents the standard deviation of the MAPE.

| | 1 Week | | | | | |
|---|---|---|---|---|---|---|
| **State** | **tf-v-SIR** | **SLOW** | **SIMLR** | **LNQ-ens1** | **Best** | **Rank** |
| Alabama | 20(16) | 19(16) | 20(12) | 21(15) | 20(12) | 1/16 |
| Alaska | 16(13) | 18(15) | 17(15) | 18(10) | 15(14) | 4/15 |
| Arizona | 21(18) | 25(19) | 22(21) | 18(16) | 18(16) | 3/16 |
| Arkansas | 20(18) | 21(29) | 24(29) | 19(19) | 19(19) | 13/16 |
| California | 15(11) | 20(15) | 13(10) | 13(10) | 13(10) | 1/16 |
| Colorado | 15(15) | 19(11) | 16(12) | 13(8) | 13(8) | 2/16 |
| Connecticut | 17(12) | 19(10) | 17(11) | 23(17) | 17(11) | 1/16 |
| Delaware | 20(14) | 18(14) | 19(13) | 15(11) | 15(11) | 4/16 |
| Washington DC | 23(15) | 19(13) | 23(15) | 15(10) | 15(10) | 8/16 |
| Florida | 12(11) | 13(7) | 12(8) | 9(7) | 9(7) | 2/16 |
| Georgia | 16(12) | 16(13) | 16(14) | 16(15) | 16(15) | 3/16 |
| Hawaii | 27(22) | 23(15) | 25(17) | 18(13) | 18(13) | 13/15 |
| Idaho | 16(11) | 16(10) | 14(10) | 14(10) | 14(10) | 2/16 |
| Illinois | 13(12) | 17(10) | 12(9) | 12(8) | 12(9) | 1/17 |
| Indiana | 11(10) | 17(10) | 15(10) | 13(11) | 13(11) | 3/17 |
| Iowa | 23(18) | 21(15) | 22(15) | 20(22) | 20(14) | 5/16 |
| Kansas | 16(15) | 20(15) | 18(12) | 21(14) | 18(12) | 1/16 |
| Kentucky | 16(11) | 16(8) | 15(9) | 12(9) | 12(9) | 2/16 |
| Louisiana | 24(17) | 23(22) | 24(22) | 21(19) | 21(19) | 3/16 |
| Maine | 17(15) | 19(15) | 18(15) | 14(11) | 14(11) | 2/16 |

**Table A1.** *Cont.*

| State | 1 Week | | | | | |
| | tf-v-SIR | SLOW | SIMLR | LNQ-ens1 | Best | Rank |
|---|---|---|---|---|---|---|
| Maryland | 14(12) | 15(12) | 13(12) | 11(7) | 11(7) | 2/16 |
| Massachusetts | 15(10) | 16(11) | 13(9) | 14(10) | 13(9) | 1/16 |
| Michigan | 15(10) | 20(10) | 16(11) | 19(11) | 16(11) | 1/16 |
| Minnesota | 19(17) | 21(16) | 20(14) | 15(12) | 15(12) | 4/16 |
| Mississippi | 19(16) | 17(16) | 19(15) | 16(12) | 16(12) | 5/16 |
| Missouri | 20(14) | 19(13) | 21(15) | 12(38) | 11(36) | 14/16 |
| Montana | 19(17) | 21(12) | 19(15) | 35(104) | 18(13) | 2/16 |
| Nebraska | 20(18) | 20(16) | 20(15) | 18(13) | 18(13) | 5/16 |
| Nevada | 18(17) | 20(15) | 20(15) | 15(11) | 15(11) | 5/16 |
| New Hampshire | 18(14) | 18(13) | 16(14) | 17(11) | 16(14) | 1/16 |
| New Jersey | 11(10) | 13(10) | 11(9) | 14(10) | 11(9) | 1/16 |
| New Mexico | 15(10) | 20(12) | 15(11) | 15(11) | 15(11) | 2/16 |
| New York | 12(9) | 14(10) | 13(8) | 11(9) | 11(9) | 2/16 |
| North Carolina | 12(10) | 14(10) | 13(9) | 12(9) | 12(9) | 2/16 |
| North Dakota | 22(22) | 23(24) | 23(23) | 16(13) | 16(13) | 8/16 |
| Ohio | 12(9) | 16(10) | 13(10) | 11(8) | 11(8) | 2/16 |
| Oklahoma | 22(23) | 24(25) | 23(24) | 15(11) | 15(11) | 13/16 |
| Oregon | 19(13) | 18(13) | 18(13) | 13(10) | 13(10) | 4/16 |
| Pennsylvania | 13(11) | 15(12) | 15(11) | 11(8) | 11(8) | 3/17 |
| Rhode Island | 14(11) | 17(11) | 13(11) | 23(15) | 13(11) | 1/16 |
| South Carolina | 16(13) | 16(11) | 16(13) | 12(8) | 12(8) | 7/16 |
| South Dakota | 18(12) | 17(14) | 17(11) | 15(10) | 15(10) | 2/16 |
| Tennessee | 18(15) | 19(15) | 22(16) | 18(12) | 18(13) | 12/16 |
| Texas | 24(22) | 23(28) | 25(29) | 20(18) | 20(21) | 7/16 |
| Utah | 14(14) | 17(11) | 16(13) | 11(10) | 11(10) | 7/16 |
| Vermont | 25(20) | 20(15) | 21(14) | 21(15) | 21(14) | 1/16 |

**Table A2.** Comparison of MAPE between different models across all the states in the US 2 weeks in advance. The number in parenthesis represents the standard deviation of the MAPE.

| State | 2 Weeks | | | | | |
| | tf-v-SIR | SLOW | SIMLR | LNQ-ens1 | Best | Rank |
|---|---|---|---|---|---|---|
| Alabama | 32(27) | 32(30) | 32(27) | 30(19) | 30(24) | 3/16 |
| Alaska | 30(32) | 30(25) | 27(25) | 28(22) | 27(24) | 2/15 |
| Arizona | 41(32) | 46(37) | 38(36) | 32(28) | 32(28) | 4/16 |
| Arkansas | 39(56) | 40(61) | 45(61) | 32(43) | 30(28) | 14/16 |
| California | 22(20) | 41(31) | 24(21) | 25(19) | 24(21) | 1/16 |
| Colorado | 31(28) | 30(19) | 33(26) | 24(19) | 24(19) | 11/16 |
| Connecticut | 27(25) | 29(18) | 29(26) | 33(18) | 29(26) | 1/16 |
| Delaware | 26(19) | 26(19) | 26(19) | 20(16) | 20(16) | 5/16 |
| Washington DC | 34(22) | 26(16) | 34(23) | 23(13) | 23(13) | 8/16 |
| Florida | 20(14) | 22(11) | 20(13) | 14(10) | 14(10) | 3/16 |
| Georgia | 25(18) | 31(19) | 27(19) | 22(20) | 22(20) | 4/16 |
| Hawaii | 41(38) | 32(30) | 39(36) | 29(23) | 28(23) | 7/15 |

**Table A2.** *Cont.*

| | 2 Weeks | | | | | |
|---|---|---|---|---|---|---|
| **State** | **tf-v-SIR** | **SLOW** | **SIMLR** | **LNQ-ens1** | **Best** | **Rank** |
| Idaho | 25(24) | 27(20) | 24(23) | 24(16) | 24(23) | 1/16 |
| Illinois | 23(18) | 31(19) | 27(19) | 23(16) | 23(16) | 3/17 |
| Indiana | 27(21) | 31(23) | 31(23) | 24(22) | 23(16) | 13/17 |
| Iowa | 36(45) | 33(21) | 33(26) | 34(32) | 31(24) | 3/16 |
| Kansas | 32(28) | 35(29) | 33(30) | 24(17) | 24(17) | 5/16 |
| Kentucky | 26(22) | 28(14) | 25(22) | 19(15) | 19(15) | 7/16 |
| Louisiana | 31(35) | 31(39) | 31(39) | 29(24) | 29(24) | 3/16 |
| Maine | 34(28) | 31(27) | 34(30) | 23(18) | 23(18) | 6/16 |
| Maryland | 24(18) | 26(19) | 23(18) | 22(16) | 22(16) | 3/16 |
| Massachusetts | 26(18) | 28(19) | 25(19) | 24(16) | 24(16) | 2/16 |
| Michigan | 33(22) | 35(19) | 33(20) | 31(16) | 27(16) | 4/16 |
| Minnesota | 40(34) | 39(32) | 41(35) | 28(23) | 28(23) | 10/16 |
| Mississippi | 26(23) | 32(25) | 31(24) | 22(18) | 22(18) | 11/16 |
| Missouri | 32(30) | 29(26) | 31(27) | 18(41) | 13(38) | 14/16 |
| Montana | 34(29) | 35(20) | 36(28) | 30(25) | 26(18) | 13/16 |
| Nebraska | 29(22) | 32(20) | 30(20) | 27(14) | 27(14) | 3/16 |
| Nevada | 31(22) | 37(25) | 33(26) | 23(18) | 23(18) | 5/16 |
| New Hampshire | 29(23) | 32(18) | 30(24) | 28(16) | 28(16) | 2/16 |
| New Jersey | 19(14) | 23(13) | 19(14) | 25(13) | 19(14) | 1/16 |
| New Mexico | 29(23) | 36(20) | 30(24) | 25(21) | 25(21) | 4/16 |
| New York | 24(18) | 24(15) | 24(18) | 21(13) | 21(13) | 4/16 |
| North Carolina | 22(14) | 26(18) | 25(18) | 17(14) | 17(14) | 6/16 |
| North Dakota | 42(39) | 41(42) | 48(44) | 32(24) | 31(20) | 13/16 |
| Ohio | 25(22) | 30(19) | 29(24) | 20(15) | 20(15) | 10/16 |
| Oklahoma | 34(30) | 34(32) | 37(31) | 25(21) | 25(21) | 13/16 |
| Oregon | 29(24) | 28(18) | 30(24) | 18(15) | 18(15) | 10/16 |
| Pennsylvania | 29(19) | 27(16) | 31(19) | 19(14) | 19(14) | 9/17 |
| Rhode Island | 21(17) | 29(19) | 24(17) | 30(19) | 24(17) | 1/16 |
| South Carolina | 27(19) | 26(20) | 27(21) | 18(13) | 18(13) | 13/16 |
| South Dakota | 30(26) | 30(28) | 32(25) | 27(20) | 27(20) | 4/16 |
| Tennessee | 30(24) | 29(26) | 34(27) | 24(19) | 24(19) | 12/16 |
| Texas | 38(49) | 35(52) | 38(51) | 26(26) | 25(34) | 8/16 |
| Utah | 27(29) | 30(20) | 32(27) | 20(19) | 20(19) | 10/16 |
| Vermont | 29(24) | 26(22) | 28(25) | 29(25) | 27(23) | 3/16 |

**Table A3.** Comparison of MAPE between different models across all the states in the US 3 weeks in advance. The number in parenthesis represents the standard deviation of the MAPE.

| | 3 Weeks | | | | | |
|---|---|---|---|---|---|---|
| **State** | **tf-v-SIR** | **SLOW** | **SIMLR** | **LNQ-ens1** | **Best** | **Rank** |
| Alabama | 40(43) | 42(41) | 34(36) | 34(27) | 34(27) | 2/16 |
| Alaska | 36(41) | 37(35) | 32(35) | 39(36) | 32(35) | 1/15 |
| Arizona | 49(44) | 70(59) | 59(60) | 42(35) | 42(35) | 6/16 |
| Arkansas | 49(52) | 54(69) | 53(70) | 40(38) | 37(26) | 12/16 |
| California | 41(49) | 67(53) | 48(50) | 34(29) | 34(29) | 6/16 |

**Table A3.** *Cont.*

| | 3 Weeks | | | | | |
|---|---|---|---|---|---|---|
| **State** | **tf-v-SIR** | **SLOW** | **SIMLR** | **LNQ-ens1** | **Best** | **Rank** |
| Colorado | 50(54) | 39(26) | 39(27) | 31(27) | 31(27) | 5/16 |
| Connecticut | 38(42) | 39(24) | 40(35) | 39(21) | 39(21) | 2/16 |
| Delaware | 39(36) | 34(28) | 39(35) | 30(23) | 30(23) | 5/16 |
| Washington DC | 48(44) | 32(23) | 35(33) | 26(20) | 26(20) | 5/16 |
| Florida | 33(26) | 34(20) | 29(20) | 19(14) | 19(14) | 3/16 |
| Georgia | 41(27) | 47(26) | 39(27) | 29(22) | 29(22) | 5/16 |
| Hawaii | 64(79) | 41(38) | 54(61) | 34(28) | 34(28) | 6/15 |
| Idaho | 38(39) | 40(31) | 35(35) | 34(26) | 33(25) | 4/16 |
| Illinois | 38(29) | 40(31) | 40(28) | 33(26) | 32(21) | 5/17 |
| Indiana | 40(33) | 44(38) | 42(34) | 35(33) | 32(23) | 11/17 |
| Iowa | 45(48) | 43(34) | 42(33) | 47(41) | 41(38) | 2/16 |
| Kansas | 47(47) | 51(46) | 45(43) | 31(20) | 31(20) | 5/16 |
| Kentucky | 38(39) | 38(25) | 31(23) | 25(18) | 25(18) | 5/16 |
| Louisiana | 36(41) | 48(58) | 46(58) | 38(28) | 38(28) | 4/16 |
| Maine | 50(39) | 43(41) | 46(39) | 33(27) | 33(27) | 5/16 |
| Maryland | 34(36) | 36(33) | 37(37) | 32(25) | 32(25) | 5/16 |
| Massachusetts | 38(34) | 40(28) | 38(30) | 33(23) | 33(23) | 2/16 |
| Michigan | 49(35) | 48(27) | 45(24) | 43(22) | 39(24) | 3/16 |
| Minnesota | 55(54) | 51(51) | 51(50) | 40(35) | 40(37) | 6/16 |
| Mississippi | 43(38) | 47(41) | 46(38) | 29(23) | 29(23) | 12/16 |
| Missouri | 36(29) | 39(39) | 39(39) | 23(47) | 19(43) | 12/16 |
| Montana | 51(46) | 42(32) | 40(34) | 40(31) | 34(21) | 7/16 |
| Nebraska | 42(33) | 44(33) | 43(33) | 37(26) | 37(26) | 4/16 |
| Nevada | 41(35) | 55(42) | 47(44) | 34(25) | 34(25) | 6/16 |
| New Hampshire | 43(38) | 42(24) | 38(22) | 34(21) | 34(21) | 3/16 |
| New Jersey | 27(24) | 31(20) | 25(17) | 34(16) | 25(17) | 1/16 |
| New Mexico | 46(47) | 52(29) | 42(32) | 33(32) | 33(32) | 8/16 |
| New York | 37(35) | 33(18) | 30(28) | 29(17) | 29(17) | 2/16 |
| North Carolina | 32(21) | 36(28) | 32(24) | 22(15) | 22(15) | 4/16 |
| North Dakota | 61(67) | 61(54) | 66(67) | 50(36) | 45(28) | 12/16 |
| Ohio | 43(42) | 41(31) | 38(31) | 28(19) | 28(19) | 5/16 |
| Oklahoma | 51(50) | 46(47) | 49(48) | 33(22) | 33(22) | 12/16 |
| Oregon | 47(49) | 39(23) | 35(26) | 28(21) | 28(21) | 2/16 |
| Pennsylvania | 46(40) | 37(23) | 38(23) | 27(17) | 27(17) | 5/17 |
| Rhode Island | 27(26) | 37(31) | 32(28) | 39(21) | 32(28) | 1/16 |
| South Carolina | 36(25) | 35(28) | 33(28) | 23(15) | 23(15) | 3/16 |
| South Dakota | 44(41) | 47(40) | 43(40) | 40(29) | 40(29) | 3/16 |
| Tennessee | 34(29) | 40(35) | 38(37) | 31(25) | 31(25) | 3/16 |
| Texas | 52(54) | 48(55) | 44(55) | 31(27) | 31(27) | 6/16 |
| Utah | 42(44) | 43(30) | 46(33) | 32(24) | 32(24) | 10/16 |
| Vermont | 44(29) | 37(21) | 41(28) | 39(24) | 38(24) | 3/16 |

**Table A4.** Comparison of MAPE between different models across all the states in the US 4 weeks in advance. The number in parenthesis represents the standard deviation of the MAPE.

| | 4 Weeks | | | | | |
|---|---|---|---|---|---|---|
| **State** | **tf-v-SIR** | **SLOW** | **SIMLR** | **LNQ-ens1** | **Best** | **Rank** |
| Alabama | 54(48) | 56(52) | 51(46) | 40(27) | 40(27) | 3/16 |
| Alaska | 58(66) | 50(36) | 47(38) | 49(32) | 46(36) | 2/15 |
| Arizona | 70(80) | 104(103) | 93(102) | 65(68) | 61(64) | 7/16 |
| Arkansas | 59(56) | 68(86) | 66(87) | 46(53) | 45(49) | 8/16 |
| California | 64(87) | 95(83) | 81(84) | 50(47) | 50(47) | 7/16 |
| Colorado | 73(90) | 52(26) | 52(33) | 41(36) | 39(24) | 6/16 |
| Connecticut | 60(65) | 46(34) | 58(49) | 44(23) | 44(23) | 6/16 |
| Delaware | 44(47) | 39(37) | 46(41) | 34(34) | 34(34) | 5/16 |
| Washington DC | 65(64) | 37(30) | 46(48) | 35(34) | 35(34) | 5/16 |
| Florida | 47(46) | 52(42) | 47(45) | 27(26) | 27(26) | 5/16 |
| Georgia | 48(40) | 64(35) | 59(33) | 38(32) | 38(32) | 7/16 |
| Hawaii | 102(153) | 55(43) | 77(98) | 45(43) | 45(43) | 6/15 |
| Idaho | 55(53) | 54(41) | 53(44) | 41(40) | 41(40) | 7/16 |
| Illinois | 53(38) | 51(43) | 54(40) | 43(37) | 39(27) | 5/17 |
| Indiana | 56(51) | 61(55) | 56(54) | 45(49) | 44(33) | 6/17 |
| Iowa | 61(72) | 55(46) | 53(46) | 55(56) | 50(45) | 2/16 |
| Kansas | 66(68) | 68(69) | 59(58) | 43(26) | 43(26) | 5/16 |
| Kentucky | 50(49) | 47(39) | 43(36) | 34(23) | 34(23) | 5/16 |
| Louisiana | 48(49) | 68(66) | 64(67) | 44(35) | 44(35) | 7/16 |
| Maine | 69(64) | 56(55) | 62(59) | 43(40) | 43(40) | 6/16 |
| Maryland | 51(71) | 45(45) | 53(61) | 42(43) | 42(43) | 6/16 |
| Massachusetts | 49(52) | 50(40) | 47(45) | 45(38) | 45(38) | 2/16 |
| Michigan | 62(67) | 56(36) | 51(37) | 53(32) | 51(44) | 2/16 |
| Minnesota | 74(87) | 65(61) | 64(62) | 55(51) | 47(41) | 5/16 |
| Mississippi | 52(49) | 62(52) | 59(51) | 38(43) | 38(43) | 6/16 |
| Missouri | 48(45) | 54(57) | 54(57) | 32(47) | 28(44) | 9/16 |
| Montana | 70(72) | 53(40) | 55(39) | 52(48) | 42(36) | 9/16 |
| Nebraska | 53(43) | 57(46) | 56(45) | 47(31) | 47(35) | 5/16 |
| Nevada | 67(54) | 77(61) | 71(65) | 41(43) | 41(43) | 8/16 |
| New Hampshire | 52(50) | 50(30) | 43(33) | 40(25) | 40(25) | 2/16 |
| New Jersey | 45(62) | 36(24) | 40(54) | 43(24) | 38(23) | 3/16 |
| New Mexico | 69(80) | 73(39) | 65(48) | 46(48) | 45(29) | 7/16 |
| New York | 48(43) | 41(22) | 39(31) | 37(25) | 33(22) | 4/16 |
| North Carolina | 41(33) | 48(41) | 45(36) | 29(22) | 29(22) | 6/16 |
| North Dakota | 79(112) | 83(77) | 94(93) | 72(63) | 60(53) | 9/16 |
| Ohio | 60(58) | 54(44) | 52(44) | 35(31) | 35(31) | 6/16 |
| Oklahoma | 81(92) | 61(67) | 70(81) | 42(32) | 42(40) | 9/16 |
| Oregon | 63(65) | 49(33) | 43(35) | 39(27) | 39(27) | 2/16 |
| Pennsylvania | 63(54) | 47(29) | 47(30) | 35(27) | 35(27) | 5/17 |
| Rhode Island | 37(39) | 46(45) | 42(43) | 44(27) | 42(43) | 1/16 |
| South Carolina | 45(31) | 49(37) | 49(36) | 28(21) | 28(21) | 8/16 |
| South Dakota | 51(47) | 64(46) | 62(45) | 54(40) | 52(30) | 9/16 |
| Tennessee | 48(48) | 58(49) | 58(50) | 43(31) | 43(31) | 5/16 |

**Table A4.** *Cont.*

| | 4 Weeks | | | | | |
|---|---|---|---|---|---|---|
| **State** | **tf-v-SIR** | **SLOW** | **SIMLR** | **LNQ-ens1** | **Best** | **Rank** |
| Texas | 63(62) | 59(67) | 58(67) | 37(34) | 37(34) | 6/16 |
| Utah | 55(70) | 56(44) | 58(50) | 40(36) | 40(36) | 7/16 |
| Vermont | 56(67) | 41(26) | 49(55) | 45(27) | 41(26) | 4/16 |

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
