# Peer review of "SIMLR: Machine Learning inside the SIR Model for COVID-19 Forecasting"

_forecasting, doi:10.3390/forecast4010005_

Round 1
Reviewer 1 Report
- Equation 3 contains a lot of variables. They are defined in different locations in the manuscript. You need to define them in one place. I suggest adding them immediately after equation 3.
- Figure 2 can be re-designed in a clearer way. It is confusing for the readers.
- Equation 5 needs more explanation, what do you mean by "arg"
- It would be great if you can omit one or two equations to reduce the ambiguity and confusion
- MAPE and MAE are not enough to measure the model's manifestation. You need to reduce some graphs of MAPE and MAE, and add the correlation coefficient instead. You can just show one example of CC, will be enough
- The conclusion section is missing, I prefer to have one for readers who don't have a lot of time to go through each section of your paper.
Reviewer 2 Report
1) Interesting paper
2) Writing: well-written and argued. Few grammatical errors and typos can be fixed (during the revision time).
3) Motivation: good; but can be extended. Suggest to address whether the model can be used for long-term (or short-term) prediction.
4) Experiments: good
5) Take-home-messages, at times, are not clear.
6) References (see below) can help enrich the paper:
+ COVID-19 Prediction Models and Unexploited Data. J Med Syst 44, 170 (2020). https://doi.org/10.1007/s10916-020-01645-z
+ Revisited COVID-19 Mortality and Recovery Rates: Are we Missing Recovery Time Period?. J Med Syst44, 202 (2020). https://doi.org/10.1007/s10916-020-01668-6
+ Centers for Disease Control and Preventions (CDC). “COVID-19 Mathematical Modeling” Source: National Center for Immunization and Respiratory Diseases (NCIRD), Division of Viral Diseases (May 26, 2020) URL: https://www.cdc.gov/coronavirus/2019-ncov/covid-data/mathematical-modeling.html
+ https://doi.org/10.1016/j.ijforecast.2020.08.004
+ https://www.nejm.org/doi/10.1056/NEJMp2016822
Reviewer 3 Report
In my opinion, the work has been done at a high level and is aimed at solving an urgent problem of current time - forecasting the development of the COVID-19 epidemic.
The approach of the authors is interesting - it combines dynamic SIR model and ML techniques to make more accurate predictions. It of course should be noted that all these forecasts may be done under existing data about infections (cases) and deaths. And the data is limited by testing capabilities and more other factors.
The methods and methodology are well described, the results are clearly presented. They are demonstrated the effectiveness of the proposed approach.
I do not have any remarks to the methodology of approach in general. In my opinion is that the paper may be published in Forecasting.
Author Response
Thanks a lot for the comments about our manuscript. We hope that this paper will motivate people to combine traditional epidemiological models with machine learning techniques to produce effective predictors.
Round 2
Reviewer 2 Report
well-revised.